# GENERATIVE GRADUAL DOMAIN ADAPTATION WITH OPTIMAL TRANSPORT

## ABSTRACT

Unsupervised domain adaptation (UDA) adapts a model from a labeled source domain to an unlabeled target domain in a one-off way. Though widely applied, UDA faces a great challenge whenever the distribution shift between the source and the target is large. Gradual domain adaptation (GDA) mitigates this limitation by using intermediate domains to gradually adapt from the source to the target domain. However, it remains an open problem on how to leverage this paradigm when the given intermediate domains are missing or scarce. To approach this practical challenge, we propose **G**enerative Gradual D**O**main **A**daptation with Optimal **T**ransport (GOAT), an algorithmic framework that can generate intermediate domains in a data-dependent way. More concretely, we first generate intermediate domains along the Wasserstein geodesic between two given consecutive domains in a feature space, and apply gradual self-training, a standard GDA algorithm, to adapt the source-trained classifier to the target along the sequence of intermediate domains. Empirically, we demonstrate that our GOAT framework can improve the performance of standard GDA when the given intermediate domains are scarce, significantly broadening the real-world application scenarios of GDA.

## 1 INTRODUCTION

Modern machine learning models suffer from data distribution shifts across various settings and datasets [Gulrajani & Lopez-Paz, 2021; Sagawa et al., 2021; Koh et al., 2021; Hendrycks et al., 2021; Wiles et al., 2022], i.e., trained models may face a significant performance degrade when the test data come from a distribution largely shifted from the training data distribution. Unsupervised domain adaptation (UDA) is a promising approach to address the distribution shift problem by adapting models from the training distribution (source domain) with labeled data to the test distribution (target domain) with unlabeled data [Ganin et al., 2016; Long et al., 2015; Zhao et al., 2018; Tzeng et al., 2017]. Typical UDA approaches include adversarial training [Ajakan et al., 2014; Ganin et al., 2016], distribution matching [Zhang et al., 2019; Tachet des Combes et al., 2020], optimal transport [Courty et al., 2016; 2017], and self-training (aka pseudo-labeling) [Liang et al., 2019; 2020; Zou et al., 2018; 2019]. However, as the distribution shifts become large, these UDA algorithms have been observed to suffer from significant performance degradation [Kumar et al., 2020; Sagawa et al., 2021; Abnar et al., 2021; Wang et al., 2022a]. This empirical observation is consistent with theoretical analyses [Ben-David et al., 2010; Zhao et al., 2019], which indicate that the expected test accuracy of a trained model in the target domain degrades as the distribution shift becomes larger.

To tackle a large data distribution shift, one may resort to an intuitive divide-and-conquer strategy, e.g., breaking the large shift into pieces of smaller shifts and resolving each piece with a classical UDA approach. Concretely, the data distribution shift between the source and target can be divided into pieces with intermediate domains bridging the two (i.e., the source and target). In settings, e.g., gradual domain adaptation (GDA) [Kumar et al., 2020; Abnar et al., 2021; Chen & Chao, 2021; Wang et al., 2022a; Gadermayr et al., 2018; Wang et al., 2020; Bobu et al., 2018; Wulfmeier et al., 2018], where the intermediate domains with unlabeled data are available to the learner, Kumar et al. [2020] proposed a simple yet effective algorithm, Gradual Self-Training (GST), which applies self-training consecutively along the sequence of intermediate domains towards the target. For GST, Kumar et al. [2020] proved an upper bound on the target error of gradual self-training, and Wang et al. [2022a] provided a significantly improved bound under relaxed assumptions, corroborating the effectiveness of gradual self-training in reducing the target domain error with unlabeled intermediate domains.

Figure 1: A schematic diagram comparing Direct Unsupervised Domain Adaptation (UDA) vs. Gradual Domain Adaptation (GDA), using an example of Rotated MNIST (60° rotation).

However, although the unlabeled intermediate domains have been shown to be useful for domain adaptation both empirically and theoretically, they are scarce in many real-world applications, hindering the wide deployment of GDA methods to overcome large distribution shifts. Hence, it is natural to ask, in cases of insufficient intermediate domains, is it possible to generate more intermediate domains useful for gradual domain adaptation? How should we generate these intermediate domains?

In this paper, we provide an affirmative answer to the first question along with an effective solution to the second, which we term Generative Gradual Domain Adaptation with Optimal Transport (GOAT). Our solution for intermediate domain generation is inspired by a theoretical insight from the generalization error of gradual self-training in Wang et al. [2022a]:

> *In order to minimize the generalization error on the target domain, the sequence of intermediate domains should be placed uniformly along the Wasserstein geodesic between the source and target domains.*

At a high-level, GOAT contains the following steps: i) generate intermediate domains between each pair of consecutive given domains along the Wasserstein geodesic in a feature space, ii) apply gradual self-training (GST) [Kumar et al., 2020], a popular GDA algorithm, over the sequence of given and generated domains.

Empirically, we conduct experiments on Rotated MNIST and Portraits [Ginosar et al., 2015], two benchmark datasets commonly used in the literature of GDA. The experimental results show that our GOAT significantly outperforms vanilla GDA, especially when the number of given intermediate domains is small. The empirical results also confirm the theoretical insights from Wang et al. [2022a]: 1). when the distribution shift between a pair of consecutive domains is small, one can generate more intermediate domains to further improve the performance of GDA; 2). there exists an optimal choice for the number of generated intermediate domains.

## 2 PRELIMINARIES

**Notation** We refer to the input and output space as $\mathcal{X}, \mathcal{Y}$, respectively, and we use $X, Y$ to represent random variables that take values in $\mathcal{X}, \mathcal{Y}$. A domain corresponds to an underlying data distribution $\mu(X, Y)$ over $\mathcal{X} \times \mathcal{Y}$. As we focus on unlabeled samples, we use $\mu(X)$ to denote the marginal distribution of $X$. Classifiers from the hypothesis class $\mathcal{H} : \mathcal{X} \mapsto \mathcal{Y}$ are trained with the loss function $\ell(\cdot, \cdot)$. We use $\mathbf{1}_n$ to refer to a $n$-dimension vector with all 1s. $\mathbb{1}[\cdot]$ stands for the indicator function.

**Unsupervised Domain Adaptation (UDA)** In UDA, we have a source domain and a target domain. During the training stage, the learner can access $m$ labeled samples from the source domain and $n$ unlabeled samples from the target domain. In the test stage, the trained model will then be evaluated by its prediction accuracy on samples from the target domain.

**Gradual Domain Adaptation (GDA)** Most UDA algorithms adapt models from the source to target in a one-step fashion, which can be challenging when the distribution shift between the two is large. Instead, in the setting of GDA, there exists a sequence of additional $T - 1$ unlabeled intermediate domains bridging the source and target. We denote the underlying data distributions of these intermediate domains as $\mu_1(X, Y), \dots, \mu_{T-1}(X, Y)$, with $\mu_0(X, Y)$ and $\mu_T(X, Y)$ being the source and target domains, respectively. In this case, for each domain $t \in \{1, \dots, T\}$, the learner has access to $S_t$, a set of $n$ unlabeled data drawn i.i.d. from $\mu_t(X)$. Same as UDA, the goal of GDA is still to make accurate predictions on test data from the target domain, while the learner can train over $m$ labeled source data and $nT$ unlabeled data from $\{S_t\}_{t=1}^{T}$. To contrast the setting of UDA and

GDA, we provide an illustration in Fig. 1 that compares UDA with GDA, using the Rotated MNIST dataset as an example.

**Gradual Self-Training (GST)** When there are multiple intermediate domains available to the learner, one classic algorithm for GDA is gradual self-training: first, train a classifier on labeled source data with the empirical risk minimization (ERM) principle to obtain a source-trained classifier $h_0 \in \mathcal{H}$. Then, fit it to the set of unlabeled target data, $S_1$, with self-training to obtain the classifier $h_1$:

$$h_1 = \text{ST}(h_0, S_1) = \underset{f \in \mathcal{H}}{\arg\min} \sum_{x \in S_1} \ell(f(x), h_0(x)), \tag{1}$$

where ST denotes the process of self-training and $h_0(x)$ represents the pseudo-label provided by $h_0$ on each unlabeled target sample $x$. The above process will then be repeatedly applied to each pair of consecutive domains $(\mu_t, \mu_{t+1})$, until one obtains a final classifier for the target domain $\mu_T$. Note that in this process, at each stage $t$, the classifier $h_{t-1}$ from the last stage will be used to generate pseudo-labels for the unlabeled data $S_t$, hence the name gradual self-training.

Intuitively, one can expect that when the distribution shift between each consecutive pair of intermediate domains is large, the quality of the pseudo-labels obtained from the previous classifier can degrade significantly, hence hurting the final target generalization. This scenario is particularly relevant when the number of given intermediate domains is relatively small.

**Theoretical Guarantees** Kumar et al. [2020] provided the first target domain error bound for gradual self-training under certain assumptions, theoretically justifying the effectiveness of GST. However, the error bound of Kumar et al. [2020] grows exponentially in $T$ (the number of domains), which contradicts empirical observations that the optimal number of intermediate domains is relatively large over multiple GDA datasets [Abnar et al., 2021; Chen & Chao, 2021; Wang et al., 2022a]. To resolve this issue, Wang et al. [2022a] proved a significantly improved error bound for GST under weaker assumptions than those of Kumar et al. [2020], expressed as

$$\varepsilon_T(h_T) \leq \varepsilon_0 + \tilde{\mathcal{O}}\left(T\Delta + \frac{T}{\sqrt{n}} + \frac{1}{\sqrt{nT}}\right), \tag{2}$$

where $\varepsilon_t(h)$ stands for the population loss of classifier $h \in \mathcal{H}$ in domain $t$ (i.e., $\varepsilon_t(h) \equiv \varepsilon_{\mu_t}(h) \triangleq \mathbb{E}_{x,y \sim \mu_t}[\ell(h(x), y)])$ and $\Delta$ is the average $p$-Wasserstein distance between consecutive domains, i.e.,

$$\Delta = \frac{1}{T}\sum_{t=1}^{T} W_p\left(\mu_{t-1}(X,Y), \mu_t(X,Y)\right) \quad \text{for } p \geq 1. \tag{3}$$

Note that in GDA, we cannot directly measure $\Delta$ since it requires access to the joint distributions of the intermediate domains, whereas only unlabeled data are available to us. In order to bridge the gap, in this paper, we make the following assumption of the intermediate domains: there exists a feature space $\mathcal{Z}$ such that the covariate shift assumption holds over $\mathcal{Z}$, i.e., the conditional distribution of $Y$ given the feature $Z$ is invariant across all the intermediate domains. Note that this assumption is also consistent with a line of recent works under the principle of invariant risk minimization [Arjovsky et al., 2019; Rosenfeld et al., 2020; Wang et al., 2022b] that seek to find features from the inputs such that the covariate shift assumption holds. Under this assumption, the Wasserstein distance between the joint distance $W_p(\mu_{t-1}(Z,Y), \mu_t(Z,Y))$ reduces to the one between the marginal feature distribution $W_p(\mu_{t-1}(Z), \mu_t(Z))$.

## 3 GENERATIVE GRADUAL DOMAIN ADAPTATION WITH OPTIMAL TRANSPORT

Inspired by the theoretical results in (2), in this section, we shall present our algorithm to automatically generate a series of intermediate domains between any pair of consecutive given domains, with the hope that when applied to the sequence of generated intermediate domains, GST could lead to better target generalization. Before presenting the proposed algorithm, we first formally introduce several notions that will be used in the design of our algorithm. First, given two distributions, one can transform one to another with a push-forward operator.

**Definition 1** (Push-forward Operator). *Consider a continuous map $\mathcal{T} : \mathcal{X} \mapsto \mathcal{X}$ and arbitrary probability measures $\mu, \nu$ on $\mathcal{X}$. When $T$ pushes $\mu$ forward to $\nu$, we denote it with $\mathcal{T}_{\#}\mu = \nu$, which means that for any measurable set $A \subset \mathcal{X}$,*

$$\nu(A) = \mu(x \in \mathcal{X} : \mathcal{T}(x) \in A) = \mu\left(\mathcal{T}^{-1}(A)\right). \tag{4}$$

Monge [1781] formalized the optimal transport problem in 1781, which is a problem to find the push-forward operator that minimizes a total transport cost.

**Definition 2** (Optimal Transport). *Given measures $\mu, \nu$ over $\mathcal{X}$ and a cost function $c : \mathcal{X} \times \mathcal{X} \mapsto [0, \infty)$, the optimal transport map $T^*$ is the one that attains the infimum of the total transport cost:*

$$\inf_{\mathcal{T}:\mathcal{X} \mapsto \mathcal{X}} \left\{ \int_{\mathcal{X}} c(x, \mathcal{T}(x)) d\mu(x) \,\middle|\, \mathcal{T}_{\#}\mu = \nu \right\}. \tag{5}$$

One can create a path of measures that interpolates the given two, and the theory of optimal transport can help us find the optimal path that minimizes the path length measured with the Wasserstein metric, i.e., the sum of Wasserstein distance between pairs of consecutive measures along the path. This optimal path is termed the Wasserstein geodesic, which is formally defined below.

**Definition 3** (Wasserstein Geodesic). *Given two measures $\nu_0, \nu_1$ over $\mathcal{X}$ a optimal transport map $\mathcal{T}^* : \mathcal{X} \mapsto \mathcal{X}$ such that $\mathcal{T}^*_{\#}\nu_0 = \nu_1$, the (constant-speed) Wasserstein geodesic between them under Euclidean metric can be defined by the path $\mathcal{P}(\nu_0, \nu_1) := \{\nu_\lambda : 0 < \lambda < 1\}$, where $\nu_\lambda = ((1 - \lambda)\mathbf{Id} + \lambda\mathcal{T}^*)_{\#}\nu_0$, and $\mathbf{Id}$ is the identity mapping.*

### 3.1 MOTIVATIONS

The target domain error bound of gradual self-training proved by Wang et al. [2022a], i.e., Eq. (2), has a dominant term $T\Delta$, which can be interpreted as the length of the path of intermediate domains connecting the source and target. Interestingly, we find that this path is related to the **Wasserstein geodesic** between the source $\mu_0$ and target $\mu_T$, and we formalize our findings as follows.

**Proposition 1** (Path Length of Intermediate Domains). *For arbitrary intermediate domains $\mu_1, \ldots, \mu_{T-1}$, the following inequality holds,*

$$T\Delta = \sum_{t=1}^{T} W_p(\mu_{t-1}, \mu_t) \geq W_p(\mu_0, \mu_T), \tag{6}$$

*where the equality is obtained as the intermediate domains $\mu_1, \ldots, \mu_{T-1}$ sequentially fall along the Wasserstein geodesic between $\mu_0$ and $\mu_T$.*

Without explicit access to the intermediate domains, gradual domain adaptation cannot be applied. Interestingly, Proposition 1 sheds light on the task of intermediate domain generation to bridge this gap: *the generated intermediate domains should fall on or close to the Wasserstein geodesic in order to minimize the path length.*

### 3.2 COMPUTATION WITH OPTIMAL TRANSPORT

From Def. 3, we know that one has to solve an optimal transport problem to generate intermediate domains along the Wasserstein geodesic. Here, we consider the problem of optimal transport between a source domain and a target domain.

**Solving Optimal Transport with Linear Programming** In unsupervised domain adaptation (UDA), the source and target domains have finite training data. Hence, we can consider the measures of the source and target to be discrete, i.e., $\mu_0$ and $\mu_T$ only have probability mass over the finite training data points. More formally, denoting the source dataset as $S_0 = \{x_{0i}\}_{i=1}^m$ and target dataset as $S_T = \{x_{Tj}\}_{i=1}^n$, the measures $\mu_0$ and $\mu_T$ can be expressed as

$$\mu_0 = \frac{1}{m}\sum_{i=1}^{m}\delta(x_{0i}), \quad \mu_T = \frac{1}{n}\sum_{j=1}^{n}\delta(x_{Tj}), \tag{7}$$

where $\delta(x)$ represents the Dirac delta distribution at $x$ [Dirac et al., 1930]. Under the discrete case, the push-forward operator $\mathcal{T}^*$ that pushes $\mu_0$ forward to $\mu_T$ can be obtained by solving a linear program [Peyré et al., 2019].

---

**Algorithm 1** Generative Gradual Domain Adaptation with Optimal Transport (GOAT)

---

**Require:** $S_0^X = \{x_{0i}\}_{i=1}^m$, $S_T^X = \{x_{Ti}\}_{i=1}^n$; Encoder $\mathcal{E}$; Source-trained classifier $h_0$

  ENCODE DATA: $S_0^Z = \{z_{0i} = \mathcal{E}(x_{0i})\}_{i \in [m]}$; $S_T^Z = \{z_{Tj} = \mathcal{E}(x_{Tj})\}_{j \in [n]}$

  OPTIMAL TRANSPORT (OT): Solve for the OT plan $\gamma^* \in \mathbb{R}_{\geq 0}^{m \times n}$ between $S_0^Z$ and $S_T^Z$

  CUTOFF: Use a cutoff threshold to keep $\mathcal{O}(n+m)$ elements of $\gamma^*$ above the threshold and zero out the rest //Only applies to the entropy-regularized version of OT

  INTERMEDIATE DOMAIN GENERATION:

  **for** $t = 1, \ldots, T$ **do**

    Initialize an empty set $S_t^Z$

    **for** each non-zero element $\gamma_{ij}^*$ of $\gamma^*$ **do**

    $z \leftarrow \frac{T-t}{T} z_{0i} + \frac{t}{T} z_{Tj}$; Add $(z, \gamma_{ij}^*)$ to $S_t$

  GRADUAL DOMAIN ADAPTATION:

  **for** $t = 1, \ldots, T$ **do**

    $h_t = \text{ST}(h_{t-1}, S_t)$ //Can also apply sample weights to losses based on $\gamma_{ij}^*$

**Output:** Target-adapted classifier $h_T$

---

**Proposition 2.** *Consider $\mu_0$ over source data $\{x_{0i}\}_{i=1}^m$ and $\mu_T$ over target data $\{x_{Tj}\}_{i=1}^n$. Given a transport cost function $c : \mathcal{X} \times \mathcal{X} \mapsto [0, \infty)$, there exists an optimal transport map $\mathcal{T}^*$, which satisfies $\mathcal{T}_\#^* \mu_0 = \mu_T$. Furthermore, for $i \in [m]$, $\mathcal{T}^*$ maps $x_{0i}$ as follows,*

$$\mathcal{T}_\#^* \delta(x_{0i}) = \sum_{j=1}^n \gamma_{ij}^* \delta(x_{Tj}), \tag{8}$$

*where $\gamma^* \in \mathbb{R}_{\geq 0}^{m \times n}$ is the optimal transport plan, a non-negative matrix of dimension $m \times n$. The plan $\gamma^*$ can be obtained by solving the following linear program,*

$$\gamma^* = \arg\min_{\gamma \in \mathbb{R}_{\geq 0}^{m \times n}} \sum_{i,j} \gamma_{i,j} c(x_{0i}, x_{Tj}) \quad s.t. \ \ \gamma \mathbf{1}_n = \frac{1}{m} \mathbf{1}_m \ and \ \gamma^T \mathbf{1}_m = \frac{1}{n} \mathbf{1}_n \tag{9}$$

**Generating Intermediate Domains with Optimal Transport** Proposition 2 demonstrates that one can use linear programming (LP) to solve the optimal transport problem between a source dataset and a target dataset. With the optimal transport plan $\gamma^*$, one can leverage Definition 3 to generate intermediate domains along the Wasserstein geodesic. Specifically, for $t = 1, \ldots, T-1$, the measure of the intermediate domain $t$ can be obtained by the following push-forward

$$\mu_t = \left( \frac{T-t}{T} \mathbf{Id} + \frac{t}{T} \mathcal{T}^* \right)_\# \mu_0 = \sum_{i,j} \gamma_{ij}^* \delta \left( \frac{T-t}{T} x_{0i} + \frac{t}{T} x_{Tj} \right) \tag{10}$$

Intuitively, $\mu_t$ can be interpreted as a discrete measure over $n_{\gamma^*}$ data points with data weights assigned by $\gamma_{ij}^*$, where $n_{\gamma^*} := \sum_{i,j} \mathbb{1}[\gamma_{ij} > 0]$ is the number non-zero entries in the matrix $\gamma^*$.

**Space Complexity** Clearly, one needs to store the optimal transport plan matrix $\gamma^* \in \mathbb{R}_{\geq 0}^{m \times n}$, in order to generate intermediate domains with (10). Thus, the space complexity appears to be $\mathcal{O}(mn)$. However, by leveraging the theory of linear programming, one can show that the maximum number of non-zero elements of the solution $\gamma^*$ to the LP (9) is at most $m + n - 1$ [Peyré et al., 2019]. Thus, the space complexity can be reduced to $\mathcal{O}(m + n)$ when using a sparse matrix format to store $\gamma^*$.

**Time Complexity** For simplicity, let us consider $m = n$. Then, the current time complexity of solving the LP (9) is known to be $O(n^3 \log(n))$ [Cuturi, 2013; Pele & Werman, 2009].

### 3.3 PROPOSED ALGORITHM

We present our proposed algorithm in Algorithm 1. Notice that Algorithm 1 directly generates intermediate domains between the source and target domains. However, in practice, there might be a few given intermediate domains that can be used by GDA. In this case, one can simply treat each

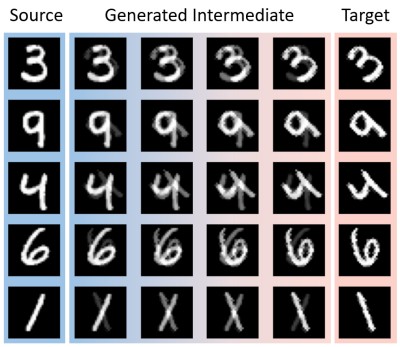

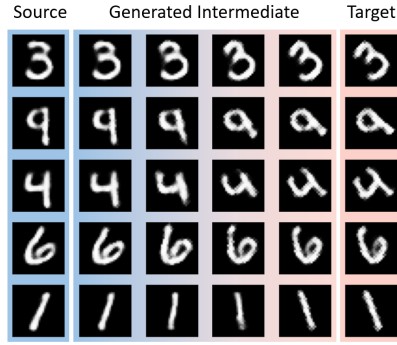

|  |  |
|---|---|
| (a) Generation in the input space. | (b) Generation in the feature space. |

Figure 2: Random samples from the generated intermediate domains.

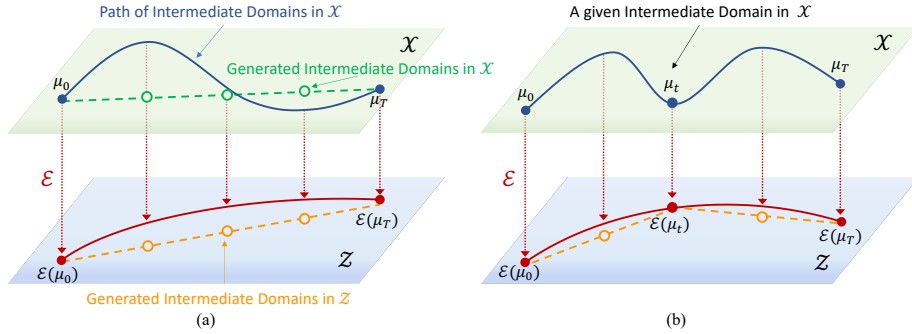

Figure 3: Illustration of the intermediate domain generation in GOAT (a) without any given intermediate domain, (b) with one given intermediate domain.

pair of consecutive domains as a source-target domain pair in a sub-level, and apply Algorithm 1 iteratively to the pairs of consecutive given domains from the source to target.

In the following, we explain some keys to our algorithm design.

**Fast Computation of Optimal Transport (OT)** The super-cubic time complexity of solving the LP in (9) essentially prevents this optimal transport approach from being scaled up to large datasets. To remedy this issue, we propose to solve an approximate objective of the OT problem (9) when it takes too long to solve the original OT exactly. Specifically, we add an entropy regularization term to the objective (9), turning it to be strictly convex, and the time complexity of solving this regularized objective is reduced to nearly $\mathcal{O}(n^2)$ from the original $\mathcal{O}(n^3 \log n)$ [Cuturi, 2013]. However, the solution to this regularized objective, i.e., the OT plan $\gamma^*$, is not guaranteed to have at most $n + m - 1$ elements anymore. Thus, the space complexity increases to $\mathcal{O}(mn)$ from $\mathcal{O}(m + n)$. We design a cutoff trick to zero out entries of tiny magnitude in $\gamma^*$ (see details in Algo. 1), reducing the space complexity back to $\mathcal{O}(m + n)$. More details regarding this part are provided in Appendix B.

**Intermediate Domain Generation in a Feature Space** The intermediate domain generation approach described above directly generates data in the input space $\mathcal{X}$. However, the generation does not have to be restricted to the input space. One can show that with a Lipschitz continuous encoder $\mathcal{E} : \mathcal{X} \mapsto \mathcal{Z}$ mapping inputs to the feature space $\mathcal{Z}$ (i.e., $z \leftarrow \mathcal{E}(x)$ for any input $x$), the order of the generation bound (2) stays the same[1] (the proof is provided in Appendix A).

*Feature Space vs. Input Space.* We use an encoder by default in Algorithm 1, since we empirically observe that directly generating intermediate domains in the input space is usually sub-optimal (see 4a for a detailed analysis). To give the readers an intuitive understanding, we provide a demo of Rotated MNIST in Fig. 2: if we apply the intermediate domain generation of Algorithm 1 in the input space, the generated data do not approximate the digit rotation well; when applying the algorithm in the latent space of a VAE (fitted to the source and target data), the generated data (obtained by

---

[1]Some terms in the bound get multiplied by a factor of the Lipschitz constant of $\mathcal{E}$.

Table 1: Accuracy (%) on Rotated MNIST.

| # Given Domains | 0 (GST) | # Generated Domains of GOAT | | | |
|---|---|---|---|---|---|
| | | 1 | 2 | 3 | 4 |
| 0 | **50.3**±**0.7** | 48.5±2.2 | 47.2±1.7 | 48.2±2.7 | 47.5±2.8 |
| 1 | 56.3±1.9 | 55.2±2.6 | 54.6±1.6 | **57.1**±**2.2** | 56.2±1.9 |
| 2 | 61.6±2.1 | 68.0±1.4 | 67.0±2.2 | 68.1±2.2 | **70.3**±**2.4** |
| 3 | 66.3±2.0 | 74.0±1.1 | **74.4**±**1.8** | 73.2±2.0 | 74.0±2.3 |
| 4 | 75.5±2.0 | 83.8±2.0 | 84.0±1.6 | **86.4**±**2.0** | 82.7±1.8 |

Table 2: Accuracy (%) on Portraits.

| # Given Domains | 0 (GST) | # Generated Domains of GOAT | | | |
|---|---|---|---|---|---|
| | | 1 | 2 | 3 | 4 |
| 0 | 73.3±1.3 | 74.0±1.3 | 73.5±2.2 | 73.6±2.5 | **74.2**±**2.5** |
| 1 | 74.5±1.6 | 76.4±1.3 | 75.5±2.6 | **76.8**±**1.5** | 74.7±1.7 |
| 2 | 77.0±1.3 | 77.4±2.1 | 79.4±2.4 | **79.9**±**1.2** | 77.2±0.9 |
| 3 | 80.7±2.3 | 80.9±1.6 | 81.8±1.3 | **82.3**±**1.3** | 81.3±1.5 |
| 4 | 82.0±1.4 | 82.8±1.5 | **83.6**±**1.5** | 82.4±1.4 | 81.8±1.6 |

the decoder of the VAE) captures the digit rotation accurately. Fig. 3a explains this superiority of the feature space over the input space with a schematic diagram: the input-space Wasserstein geodesic may not approximate the ground-truth distribution shift (e.g., rotation) due to the linearity of optimal transport; with a proper encoder $\mathcal{E}$, the feature-space Wasserstein geodesic can capture the distribution shift more accurately.

*Leveraging Given Intermediate Domain(s).* With a given intermediate domain, we generate intermediate domains with GOAT between the two pairs of consecutive domains, respectively, Fig. 3b shows that this approach can make the generated domains closer to the ground-truth path of distribution shift, explaining why GOAT benefits off given intermediate domains.

**Gradual Domain Adaptation (GDA) on Generated Intermediate Domains** With generated data of intermediate domains, one can run a GDA algorithm consecutively over the source-intermediate-target domains in the feature space. As for the choice of GDA algorithm, we adopt Gradual Self-Training (GST) [Kumar et al., 2020], since it is simple, popular, and powerful. Nevertheless, one can freely apply any other GDA algorithm on top of the generated domains.

## 4 EXPERIMENTS

Our goal of the experiment is to demonstrate the performance gain of training on generated intermediate domains in addition to given domains. We compare our method with gradual self-training [Kumar et al., 2020], which only self-trains a model along the sequence of given domains iteratively. In Sec. 4.4, we further analyze the choices of encoder $\mathcal{E}$ and transport plan $\gamma^*$ used by Algorithm 1. More details of our experiments are provided in Appendix C.

### 4.1 DATASETS

**Rotated MNIST** A semi-synthetic dataset built on the MNIST dataset [LeCun & Cortes, 1998], with 50,000 images as the source domain and the same 50,000 images rotated by 45 degrees as the target domain. Intermediate domains are evenly distributed between the source and target.

**Portraits [Ginosar et al., 2015]** A real-world gender classification dataset consisting of portraits of high school seniors from 1905 to 2013. Following Kumar et al. [2020], the dataset is sorted chronologically and split into a source domain (first 2000 images), 7 intermediate domains (next 14000 images), and a target domain (last 2000 images).

### 4.2 IMPLEMENTATION

Our code is developed in PyTorch [Paszke et al., 2019], and our experiments are run on NVIDIA RTX A6000 GPUs. We use a convolutional neural network (CNN) of 4 convolutional layers of 32 channels followed by 3 fully-connected layers of 1024 hidden neurons, with ReLU activation. We also adopt common practices of Adam optimizer [Kingma & Ba, 2015], Dropout [Srivastava et al., 2014], and BatchNorm [Ioffe & Szegedy, 2015]. To calculate the optimal transport plan between the source and target, we use the Earth Mover Distance solver from [Flamary et al., 2021]. Notice that the number of generated intermediate domains is a hyperparameter, and we show the performance for 1,2,3 or 4 generated domains between each pair of consecutive given domains. Following practices [Kumar et al., 2020], in self-training, we filter out the 10% data where the model's prediction is least confident at.

As for the implementation of GOAT (Algorithm 1), we take the first two convolutional layers as the encoder $\mathcal{E}$, and treat the last layers as the classifier $h$. Sec. 4.4 explains why use this encoder.

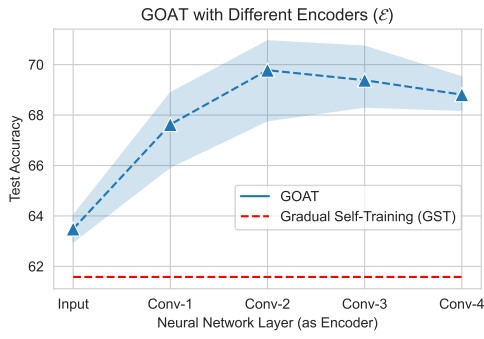 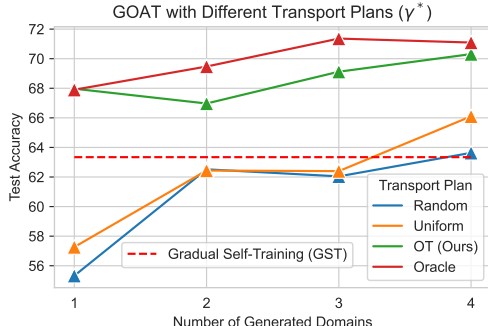

(a) Choices of the encoder ($\mathcal{E}$) in Algorithm 1.  (b) Choices of the transport plan ($\gamma^*$) in Algorithm 1.

Figure 4: Ablation studies on Rotated MNIST with 2 given intermediate domains. (a) *Different neural net layers as encoders for the intermediate domain generation of GOAT.* One can see that input space is not suitable for intermediate domain generation, and the second convolutional layer (CONV-2) is optimal. (b) *Different transport plans the intermediate domain generation of GOAT.* Obviously, our optimal transport (OT) plan significantly outperforms the baseline transport plans (Random & Uniform), and its performance is even close to the oracle.

## 4.3 EMPIRICAL RESULTS

We empirically compare our proposed GOAT with Gradual Self-Training (GST) [Kumar et al., 2020]. The results on Rotated MNIST and Portraits are shown in Table 1 and Table 2. Each experiment is run 5 times with 95% confidence interval reported. The leftmost column corresponds to the performance of GST only on given intermediate domains, which is equivalent to GOAT without any generated intermediate domain.

In Table 1 and 2, the columns ("# Generated Domains of GOAT") represent the number of *generated intermediate domains between **each pair** of consecutive given domains*, while the rows ("# Given Domains") indicate the number of *given intermediate domains*.

**Observations**  i) From the columns of both Table 1 and 2, we can observe that the performance of GOAT monotonically increases with more given intermediate domains, indicating that GOAT indeed benefits from given intermediate domains. ii) From the rows of both Table 1 and 2, we can see that with a fixed number of given domains, our GOAT can consistently outperform Gradual Self-Training (GST). The only exception is the case of Rotated MNIST without any given intermediate domain, which might be due to the challenge illustrated in Fig. 3(a).

Overall, the empirical results shown in the two tables demonstrate that our GOAT can consistently improve gradual self-training (GST) with generated intermediate domains when only a few given intermediate domains are available.

## 4.4 ABLATION STUDIES

**Choice of Encoder ($\mathcal{E}$)**  Here, we study how the choice of the encoder (i.e., feature space) affects the performance of GOAT. Since we use a CNN, we can take each network layer as the feature space. Specifically, we consider the four convolutional layers and input space as candidate choices for the encoder. Once choosing a layer, we take all layers before it (including itself) as the encoder. In this ablation study, we use Rotated MNIST dataset with 2 given intermediate domains, and let GOAT generate 4 intermediate domains between consecutive given domains. From Fig. 4a, we can observe that directly applying GOAT in the input space performs significantly worse than the optimal choice, CONV-2 (i.e., the second convolutional layer). This result justifies our use of an encoder for intermediate domain generation (instead of directly generating in the input space). Notably, Fig. 4a shows that deeper layers are not always better, showing a clear increase-then-decrease accuracy curve. Hence, we keep using CONV-2 as the encoder for GOAT in all experiments.

**Choice of Transport Plan ($\gamma^*$)**  In our Algorithm 1, the data generated along the Wasserstein geodesic are essentially linear combinations of data from the pair of given domains, with weights (for each combination) assigned by the optimal transport (OT) plan $\gamma^*$. To validate that the performance gain of GOAT indeed comes from the Wasserstein geodesic estimation instead of just linear combinations, we conduct an ablation study on GOAT in Rotated MNIST with 2 given intermediate domains. Specifically, we consider four approaches to provide the transport plan $\gamma^*$: i) a random transport plan (weights are sampled from a uniform distribution), ii) a uniform transport plan (weights are the same for all combinations), iii) the optimal transport (OT) plan provided by Algorithm 1, iv) the oracle

transport plan[2], which is the ground-truth transport plan in this study. For a fair comparison, When constructing the random and uniform plans, we ensure that the number of non-zero elements is the same as that of the oracle plan (i.e., keeping the number of generated data the same). See more details in Appendix C.

From Fig 4b, we observe that, in general, the random and uniform plans do not obtain non-trivial performance gain compared with the baseline, the vanilla Gradual Self-Training (GST) without any generated domain. In contrast, our OT plan is significantly better and achieves similar performance as the oracle, demonstrating the high quality of the OT plan and justifying our algorithm design with the Wasserstein geodesic.

## 5  RELATED WORK

**Unsupervised Domain Adaptation (UDA)**  One of the most popular approaches is invariant representation learning, which minimizes the distribution distance between the source and target domains under a certain metric in some feature space with either adversarial training [Ajakan et al., 2014; Ganin et al., 2016] or divergence minimization [Sun & Saenko, 2016; Zhang et al., 2019]. The goal of this approach is to learn representations invariant over the source and target domains, and it enjoys theoretical guarantees [Ben-David et al., 2010; Zhao et al., 2019]. Self-training (i.e., pseudo-labeling) is another popular approach for UDA [Zou et al., 2018; 2019; Liang et al., 2019; 2020], which utilizes a source-trained model to offer pseudo-labels for unlabeled target samples and further finetunes the model with these pseudo-labels. It is notable that even having achieved remarkable successes over a wide range of datasets, modern UDA approaches may still become ineffective as the distribution shifts are large enough [Kumar et al., 2020; Sagawa et al., 2021; Abnar et al., 2021].

**Gradual Domain Adaptation (GDA)**  Most UDA approaches (e.g., the ones introduced above) adapt models directly from the source to the target in a one-step style, so they are also addressed as Direct UDA [Kumar et al., 2020; Chen & Chao, 2021]. Gopalan et al. [2011] introduces the notion of intermediate domains to the area of domain adaptation. In fact, Gopalan et al. [2011] studies the standard UDA task with labeled source data and unlabeled target data, and fits PCA to the source and target data to obtain two linear representations of the two domains. They constructed the intermediate domains via linear interpolation between the two PCA subspaces, and then concatenated the linear representations of all domains to a classifier upon them. A series of follow-up works of Gopalan et al. [2011] were proposed [Zheng et al., 2012; Gong et al., 2012; Hoffman et al., 2014; Cui et al., 2014] in the following years. In the deep learning era, some empirical GDA algorithms are proposed [Gadermayr et al., 2018; Wulfmeier et al., 2018; Gong et al., 2019] in computer vision, leveraging modern deep learning techniques, but they lack theoretical guarantees. Recently, Kumar et al. [2020] proposes gradual self-training (GST), a simple yet effective GDA algorithm that enjoys theoretical guarantees, building on the self-training technique [Yarowsky, 1995; Lee et al., 2013]. Abnar et al. [2021]; Chen & Chao [2021] develop more powerful empirical GDA algorithms based on GST, and Wang et al. [2022a] significantly improves the theoretical guarantees of GST. In parallel to GST, another GDA approach is proposed [Wang et al., 2020], which also enjoys certain theoretical guarantees.

## 6  CONCLUSION

Gradual domain adaptation (GDA) attempts to address large distribution shifts between the source and target domains by adapting models along a sequence of intermediate domains. However, GDA becomes unreliable as the number of given intermediate domains is insufficient. To address this limitation of GDA, we propose a novel algorithmic framework, Generative Gradual Domain Adaptation with Optimal Transport (GOAT), which automatically generates intermediate domains along the Wasserstein geodesic (between consecutive given domains) and applies GDA on the generated domains. Our algorithm is motivated by the recent theory of Wang et al. [2022a] on gradual domain adaptation, and we develop practical computation techniques to implement the algorithm. Empirically, we show that GOAT can significantly outperform vanilla GDA when the given intermediate domains are scarce. Essentially, our GOAT is a promising framework that augments GDA with generated intermediate domains, leading GDA to be applicable to more real-world scenarios.

---

[2]The target data of the Rotated MNIST dataset are obtained by rotating training data. Thus there is a one-to-one mapping between source and target data. The oracle plan is built from the one-to-one mapping, i.e., an element $\gamma_{ij}^*$ is non-zero if and only if $x_{0i}$ is rotated to $x_{Tj}$.

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

## A  THEORETICAL ARGUMENTS

**On Proposition 1**  The inequality in (6) holds true since the Wasserstein distance metric $W_p$ is known to enjoy the property of triangle inequality. In (6), the equality is obtained as the intermediate domains $\mu_1, \ldots, \mu_{T-1}$ sequentially fall along the Wasserstein geodesic between $\mu_0$ and $\mu_T$, since the geodesic is defined as the shortest path of distributions connecting $\mu_0$ and $\mu_T$ under the $W_p$ metric.

**On Proposition 2**  This linear program (LP) formulation of optimal transport is also called Kantorovich LP in the literature. One can find details and proof of Kantorovich LP in [Peyré et al., 2019].

**On the Encoder**  With a $\rho_{\mathcal{E}}$-Lipschitz continuous encoder $\mathcal{E} : \mathcal{X} \mapsto \mathcal{Z}$ mapping inputs to the feature space $\mathcal{Z}$ (i.e., $z \leftarrow \mathcal{E}(x)$ for any input $x$), the order of the generation bound (2) stays the same. The reason is as follows: The bound (2) is linear in terms of $\rho_h$ [3], which is the Lipschitz constant of the classifier $h$; With the encoder $\mathcal{E}$, one can effectively view the whole encoder-classifier model as $f : \mathcal{X} \mapsto \mathcal{Y}$ such that $f(x) = h(\mathcal{E}(x))$; Then, the Lipschitz constant of $f$ is obviously $\rho = \rho_{\mathcal{E}} \rho_h$ since $f$ is a composite function of $h \circ \mathcal{E}$; Finally, replacing $h$ with $f$ in the analysis of Wang et al. [2022a], one can see that the order of the bound (2) stays the same, with some terms getting multiplied by a factor of $\rho_{\mathcal{E}}$ (i.e., equivalent to replacing the term $\rho_h$ with $\rho = \rho_{\mathcal{E}} \rho_h$ in the bound).

## B  MORE DETAILS ON THE PROPOSED ALGORITHM

To reduce the $\mathcal{O}(n^3 \log n)$ complexity of the exact OT calculation to $\mathcal{O}(n^2)$, we can solve the entropy-regularized OT problem Cuturi [2013] instead. Consider source data $\{x_{0i}\}_{i=1}^m$ and target data $\{x_{Tj}\}_{i=1}^n$, the entropy-regularized OT plan $\gamma_\lambda^*$ under the transport cost function $c$ is obtained by solving

$$\gamma_\lambda^* = \arg\min_{\gamma \in \mathbb{R}_{\geq 0}^{m \times n}} \sum_{i,j} \gamma_{i,j} c(x_{0i}, x_{Tj}) + \lambda \sum_{i,j} \gamma_{i,j} \log \gamma_{i,j},$$
$$\text{s.t. } \gamma \mathbf{1}_n = \frac{1}{m} \mathbf{1}_m \text{ and } \gamma^T \mathbf{1}_m = \frac{1}{n} \mathbf{1}_n, \tag{11}$$

where $\lambda$ is a regularization coefficient. The low computational complexity comes at the cost of a dense optimal transport plan, i.e., $\gamma_\lambda^*$ is generally a dense matrix rather than a sparse one [4]. Thus, $\mathcal{O}(mn)$ non-zero entries will be generated in $\gamma_\lambda^*$, and this quadratic space complexity becomes intractable for large datasets. To remedy this issue, we design two methods to zero out insignificant entries in $\gamma_\lambda^*$ to reduce the space complexity:

1. **Small-value cutoff.** Although the transport plan $\gamma_\lambda^*$ resulted from entropy-regularized OT is dense, most entries still have values close to 0. Those entries of tiny magnitude can be zeroed out without having a noticeable impact on the final results.

2. **Confidence cutoff.** Consider the one-hot encoded matrix of source labels $Y_0 \in \{0, 1\}^{m \times \#\text{class}}$ and the entropy-regularized OT plan $\gamma_\lambda^*$. The logits of target prediction by optimal label transport is

$$\widehat{Y}_T = \gamma_\lambda^{*T} Y_0. \tag{12}$$

Then, we can calculate a confidence score for each target prediction by the logits. Using a certain confidence threshold, the target samples that the transport plan is unconfident with can be filtered out, making the transport plan more sparse.

With proper choices of cutoff values, those methods can reduce the space complexity from $\mathcal{O}(mn)$ to $\mathcal{O}(m + n)$ without noticeable compromise on the final performance.

---

[3]The dependence on $\rho_h$ is hidden with the big-O notation in (2)

[4]As we discussed in Sec. 3.2, $\gamma^*$ has at most $n + m - 1$ non-zero entries, thus it is a sparse matrix.

# C  MORE DETAILS ON EXPERIMENTS

**Network Implementation.**    For the 4-layer CNN encoder used in experiments on Rotated MNIST and Portraits, we use convolutional layers with kernel size 3 and SAME padding. During self-training, we train on each domain for 10 epochs. Empirically, we verify that regularization techniques are important for the success of gradual self-training, including using dropout layers and early stopping.

For the VAE used to produce Fig. 2, we use 4 convolutional layers with kernel size 3 and max-pooling, followed by a fully-connected layer with 128 neurons as the encoder. For the decoder, we use four deconvolutional layers with kernel size 3 [Kingma & Welling, 2014]. We use ReLU activation for the layers. The encoder and decoder are jointly trained on data from source and target in an unsupervised manner with the Adam optimizer [Kingma & Ba, 2015] (learning rate as $10^{-4}$ and batch size as 512).

**Encoder Pretraining.**    We pretrain an encoder on the given domains. During pretraining, we use a 3-layer MLP on top of the encoder and perform self-training on the given domains. Specifically, we first fit the model on the source domain, then iteratively use the model to pseudo-label the next domain and self-train on it. After pretraining, the MLP is discarded and the encoder is fixed to provide features for the downstream tasks.

**OT ablation.**    When designing different plans, we make sure that the number of non-zero entries is equal so that in the domains generated by those plans, the amount of data is the same. For the random plan, we first initialize a zero matrix, then sample the same amount of entries as the ground-truth plan in the matrix, and fill in a weight value between 0 to 1 uniformly at random. For the uniform plan, we use the same procedure except that we fill in the same weight for each sampled entry. In the end, we normalize the matrix.

