# OpenReview forum: "Generative Gradual Domain Adaptation with Optimal Transport"
_ICLR.cc/2023/Conference — Submitted to ICLR 2023_

### Official Review · Reviewer_d4Li · 2022-10-20

**Confidence:** 4
**Correctness:** 4
**Technical Novelty And Significance:** 3
**Empirical Novelty And Significance:** 2
**Recommendation:** 8

**Clarity, Quality, Novelty And Reproducibility:**

Clarity: excellent. Really well written, provides enough background on all the methods needed
Quality: theoretically justified idea
Novelty: novel way of generating intermediate domains
Reproducibility: authors provide enough of the details to be able to reproduce

**Strength And Weaknesses:**

Strength:
- clarity: well written and easy to follow
- good idea inspired by the theoretical bounds - find the optimal intermediate domains
Weaknesses
- Experiments don't compare with UDA baselines
- Experiments are not convincing when there are no intermediate domains provided. May be more intermediate domains needed?
- The method is expensive

**Summary Of The Paper:**

This paper tackles the setting of unsupervised domain adaptation when the shift between the source and target is large and thus gradual domain adaptation is needed. They propose a gradual domain adaptation algorithm that creates intermediate domains for the gradual adaptation and then applies (standard) gradual self training on these created domains. Authors propose to create domains in extracted feature space (e.g. on the embedding level). They show that it does better than creating domains using the original input space. The idea of creating intermediate domains spans from theoretical bounds: synthetic domains minimize the average Wasserstein distance between the pairs of intermediate domains.


**Summary Of The Review:**

I read authors response and am increasing my score to accept
Overall very well written paper with somewhat weak experimental results. I am willing to be convinced otherwise if authors are able to provide additional results (please see below)

Method:
1) Is the number of domains generated a hyperparameter? Why do you go only up to 4 (or rather 4*number of intermediate domains provided)? Can you instead find the optimal value of it using the bounds in (2)? (e.g. min over T (1/T + T/sqrt(n) + 1/sqrt(nT))?
2) How many data points do you geenerate for each intermediate domain - is it the number of non zero elements in optimal transport matrix (so m+n approx, where m and n are source and target number of points?)
3) This method sounds extremely expensive. Apart from OT which you can solve approximately, you train additional T models to convergence. Can you add wall time comparison to Tables 1 and 2, I am trying to understand the bulk part of it (is it 4 times slower for 4 intermediate domains? Or more?)

Experiments:
1)  A bit surprised that only two datasets are used, but seems that a lot of papers on GDA use those 2 datasets. Though this popular paper https://arxiv.org/pdf/2002.11361.pdf also has experiments on Gaussian synthetic data. Do you have experiment results on such data by any chance?
2) Why not to add UDA like DANN or MMD to Table 1 and 2. You seem to say that they don't make sense since the shift is large. Do they underperform the 0-self training entry?
3) is in Table 1 and 2 "0 domains" entry correspond to self training?
Looking at table 1 it seems that when there are 0 given intermediate domains, you don't imporve over the self training at all apart from one case (portrait on 4). All drastic improvements are only when there are some intermediary domains already provided. Why is this?
4) Also i do find that you report # of generated domains this way confusing - for 3 given domains and 3 generated, you actually generated 9 domains right?

Minor:
- (6) treats T delta as the length, but it was the AVERAGE (so 1/T) in (3).
- intro: performance degrade => performance degradation

---

> ### Author Response · Authors · 2022-11-12
> **Response to d4Li (1/2)**
>
> 1. **Is the number of domains generated a hyperparameter?** Yes. Wang et al. 2022 [2] prove that there **exists** an optimal number of intermediate domains, and it provides a formula for that (Eq. 20 of [2]) -- which is just taking the optimal value of $T$ using the bound in Eq. (2) that we stated in our paper. However, the formula depends on some unknown constants that are dependent on the choices of dataset and model, etc., so we cannot directly use it in practice -- we have to treat the optimal number of intermediate domains as a hyperparameter and empirically determine it.
>
> 2. **How many data points do you generate for each intermediate domain -  is it the number of non-zero elements in the optimal transport matrix?** If we use the exact optimal transport (OT) solver, it is guaranteed that the OT solution (a $m\times n$ non-negative matrix) has exactly $m+n-1$ non-zero elements, so we have $m+n-1$ generated data in each intermediate domain. However, the exact OT solver is too costly, so we adopt an entropy-regularized OT solver (i.e., Sinkhorn solver [3]), which has more non-zero elements (still all non-negative). In our implementation with the Sinkhorn solver, we apply a cutoff method (stated in Appendix B) to keep the top $O(m+n)$ elements of the OT solution and zero out the rest.
>
> 3. **Wall time comparison to Tables 1 and 2.** We include the wall time in the tables below. The time recorded is in seconds. Notice that, same as Table 1 & 2 in our paper, the columns represent “# Generated Domains” while the rows stand for “# Given Domains.”
>
>     **i) Rotated MNIST**
>
>     |  #Given \ #Generated  | 0        | 1        | 2        | 3        | 4        |
>     |---|----------|----------|----------|----------|----------|
>     | 0 | 650.55   | 1,095.80 | 1,307.05 | 1,376.75 | 1,498.50 |
>     | 1 | 805.05   | 1,785.15 | 2,059.20 | 2,581.75 | 2,891.40 |
>     | 2 | 957.5    | 2,503.10 | 3,223.05 | 3,733.65 | 4,304.85 |
>     | 3 | 1,136.40 | 3,274.20 | 4,166.20 | 5,141.30 | 6,168.75 |
>     | 4 | 1,329.60 | 4,188.50 | 5,659.20 | 7,020.20 | 8,675.55 |
>
>     **ii) Portraits**
>
>     |  #Given \ #Generated | 0     | 1      | 2      | 3      | 4      |
>     |---|-------|--------|--------|--------|--------|
>     | 0 | 32.82 | 47     | 54.41  | 66.99  | 71.86  |
>     | 1 | 36.56 | 71.12  | 92.69  | 113.84 | 132.46 |
>     | 2 | 44.75 | 116.85 | 130.63 | 165.4  | 200.56 |
>     | 3 | 52.58 | 115.95 | 182.61 | 192.22 | 249.11 |
>     | 4 | 65.19 | 157.73 | 215    | 283.32 | 319.08 |
>
>
>     + The time does not necessarily increase linearly with the number of generated domains because we only need to calculate OT once between each pair of adjacent domains. Please also refer to our item-7 in this response below to see how we define # generated domains.
>
>     + In each step, instead of training a new model from scratch, we only finetune the classifier part of the model from the previous step, while the encoder remains unchanged. We also perform early stopping to avoid overfitting (if the model is trained till convergence, it will overfit to the noisy pseudo-labels in the intermediate domains). Thus, the model update procedure is computationally efficient.
>
>     + The computation time of Rotated MNIST is larger than Portraits because Rotated MNIST has much more data in each (ground-truth) intermediate domain, leading to longer runtime on OT computation & model training.
>         + data size of each (ground-truth) intermediate domain in Rotated MNIST: 50,000
>         + data size of each (ground-truth) intermediate domain in Portraits: 2,000
>
> 4. **Experiments on More GDA Datasets.** We just conducted experiments in two additional datasets – see item-1 of our [common response thread](https://openreview.net/forum?id=E1_fqDe3YIC&noteId=WPibEbBSzv) above for the results.
>
> 5. **Comparison with DANN.** We conducted experiments for DANN on the portraits and Rotated MNIST dataset with no intermediate domain given (i.e., the UDA setting). For portraits, DANN shows a similar performance to self-training. However, for Rotated MNIST, the performance of DANN is noticeably worse, verifying our claim that one-off UDA algorithms may not be suitable for problems with large distribution shifts (such as Rotated MNIST).
>
>     |               | DANN         | Self-Training        |
>     |---------------|--------------|------------|
>     | Rotated MNIST | 43.9 ± 1.4 | **50.3 ± 0.7** |
>     | Portraits     | **73.7 ± 0.5** | **73.3 ± 1.3** |
>
> 6. **The Case of No Ground-Truth Intermediate Domain.** Please see our response to this question in item-3 of our [common response thread](https://openreview.net/forum?id=E1_fqDe3YIC&noteId=WPibEbBSzv) above.

---

> > ### Author Response · Authors · 2022-11-12
> > **Response to Reviewer d4Li (2/2)**
> >
> > 7. **Definition of "# generated domains".** “# given domains” in our tables means the number of ground-truth intermediate domains given in the dataset, which does not count the source and target domains. The “# generated domains” are counted between each pair of adjacent ground-truth domains (e.g., between the source domain and the first ground-truth intermediate domain, or between the $i$-th and $(i+1)$-th ground-truth intermediate domains). For instance, in the case of “# given domains = 3” and “#  generated domains = 3”, we have 5 ground-truth domains (source, target and 3 intermediate domains) and 4*3=12 generated domains (since there are 4 pairs of adjacent domains along the sequence of 5 ground-truth domains), leading to 17 domains in total.
> >
> >     We will clarify this definition more clearly in our revision of the paper.
> >
> >
> > **References**
> >
> > [1] Nicolas Courty, Rémi Flamary, Devis Tuia, and Alain Rakotomamonjy. Optimal transport for domain adaptation. IEEE transactions on pattern analysis and machine intelligence, 39(9):1853–1865, 2016.
> >
> > [2] Haoxiang Wang, Bo Li, Han Zhao. Understanding gradual domain adaptation: Improved analysis, optimal path and beyond. In ICML 2022.
> >
> > [3] Marco Cuturi. Sinkhorn Distances: Lightspeed Computation of Optimal Transport. NeurIPS 2013.

---

### Official Review · Reviewer_3XRG · 2022-10-24

**Confidence:** 4
**Correctness:** 3
**Technical Novelty And Significance:** 3
**Empirical Novelty And Significance:** 3
**Recommendation:** 3

**Clarity, Quality, Novelty And Reproducibility:**

The clarity, quality, and novelty are only marginally significant. Not sure if the paper results can be reproduced.

**Strength And Weaknesses:**

Strengths:
(1)	This paper gives a clearly theoretical analysis which explanation of the definition and generation method of the intermediate domains

Weaknesses:
(1)	Is there some problem with Equation 10. How does it become the mixup of samples? please give a detailed proof of this problem.
(2)	The experiment is insufficient. The authors mentioned [1] in related work, which also relies on intermediate domain generation, however, is compared in this work.
(3)	While the generated intermediate domains by the proposed GOAT method is demonstrate in Figure 2 for the rotated MINIST data, other more complicated scenarios are not shown.
(4)	In Table, given domains=0, GOAT does not bring improvement, why?
(5)	Some papers have also achieved very good results in constructing intermediate domain at the input level [2,3]. Discussion and comparisons are needed.
[1] Gong R, Li W, Chen Y, et al. Dlow: Domain flow for adaptation and generalization[C]//Proceedings of the IEEE/CVF conference on computer vision and pattern recognition. 2019: 2477-2486.
[2] Na J, Jung H, Chang H J, et al. Fixbi: Bridging domain spaces for unsupervised domain adaptation[C]//Proceedings of the IEEE/CVF Conference on Computer Vision and Pattern Recognition. 2021: 1094-1103.
[3] Na J, Han D, Chang H J, et al. Contrastive Vicinal Space for Unsupervised Domain Adaptation[J]. arXiv preprint arXiv:2111.13353, 2021.


**Summary Of The Paper:**

When the distribution gap between the source domain and the target domain is very large, some intermediate domains are usually used to make the model gradually adapt to the target domain. The author proposes the GOTA method to generate intermediate domains based on Wasserstein geodesic. Empirically, this GOAT framework can improve the performance of standard GDA(Gradually Domain Adaptation) when the given intermediate domains are scarce, significantly broadening the real-world application scenarios of GDA

**Summary Of The Review:**

The generation theory of the intermediate domain is interesting, but there are some issues with the derivation. In addition, the author's experiment is relatively simple and lacks justification for the existing DA datasets

---

> ### Author Response · Authors · 2022-11-12
> **Response to Reviewer 3XRG (1/2)**
>
> 1. **How does Eq (10) become the mixup of samples?** Eq. 10 is how optimal transport operates in the Euclidean space under the Euclidean metric (i.e., L2 distance metric). We provide a brief proof below step by step.
>
>     **i)** The source dataset and the target dataset, $ \\{ x_{01}, x_{02}... x_{0m} \\}$ and $ \\{ x_{T1}, x_{T2}... x_{Tn} \\}$, are viewed as two discrete measures --> the source data distribution is expressed as $\\mu_0 = \\frac{1}{m} \\sum_{i=1}^m \\delta(x_{0i})$ and the target data distribution is $\\mu_T = \\frac{1}{n} \\sum_{i=1}^n \delta (x_{Tj})$, where $\delta$ is the [Dirac delta function](https://en.wikipedia.org/wiki/Dirac_delta_function).
>
>     **ii)** For $\\mu_0$ (a discrete measure uniformly over $m$ points), a push-forward operator $\\mathcal T$ essentially just moves the position of each point $x_{0i}$ to some new position(s) with an assigned weight(s). For instance, the $\\mathcal{T}^*$ in Eq. 8 just moves each source data point $x_{0i}$ to the positions of target data points with weights assigned by $\\{\\gamma^*_{ij} \\}_{j=1}^n$, i.e.,
>
>     $$\\mathcal{T}^* {\small \\#}\delta(x_{0i}) = \\sum_{j=1}^n \\gamma_{ij}^* \delta( x_{Tj})$$
>     where $\\sum_{j=1}^n \\gamma^*_{ij} = 1$. Notice that this is Eq. 8 in the paper (Eq. 8 omits the $1/m$ constant, which we will add in the revision).
>
>     Thus, when $\mathcal T^*$ acts on $\mu_0$, it leads to
>     $$\\mathcal{T}^* {\small \\#} \mu_0  =  \\mathcal{T}^* {\small \\#} \left( \\frac{1}{m} \\sum_{i=1}^m \\delta(x_{0i}) \right)=  \frac{1}{m} \sum_{i=1}^{m} \\sum_{j=1}^n \\gamma_{ij}^*  \delta (x_{Tj})   $$
>
>     **iii)** By Definition 3, we know the push-forward induced by the Wasserstein geodesic from $\mu_0$ to $\mu_T$ is $$\mathcal T_t = \frac{T-t}{T} \mathbf{\mathrm{Id}}  + \frac{t}{T} \mathcal T^*~,$$ where $\mathbf{\mathrm{Id}}$ is the identity mapping. Thus, for each source data point $x_{0i}$,  $\mathcal T_t $ push-forwards it as follows (see Remark 7.1 of [4] for the computation rule of adding push-forward operators in the Euclidean space) $$\mathcal T_t {\small \\#}\delta(x_{0i}) = \\left( \frac{T-t}{T} \mathbf{\mathrm{Id}}  + \frac{t}{T} \mathcal T^* \\right) {\small \\#} \delta(x_{0i}) =  \sum_{j=1}^n \gamma^*_{ij} ~\delta \left( \frac{T-t}{T} x_{0i} +  \frac{t}{T }x_{Tj} \right) ~.$$
>
>     Hence, as $\mathcal T_t$ acts on $\mu_0$, it push-forwards all data points in $\mu_0$ in the above way, leading to Eq. 10: $$\mathcal {T_t} {\small \\#}  \mu_0= \left( \frac{T-t}{T} \mathbf{\mathrm{Id}}  + \frac{t}{T} \mathcal T^* \right) {\small \\#}  \mu_0 = \left( \frac{T-t}{T} \mathbf{\mathrm{Id}}  + \frac{t}{T} \mathcal T^* \right) {\small \\#}  \\frac{1}{m} \\sum_{i=1}^m \\delta(x_{0i}) =  \frac{1}{m} \sum_{i=1}^m \sum_{j=1}^n \gamma^*_{ij} ~\delta\left( \frac{T-t}{T} x_{0i} +  \frac{t}{T }x_{Tj} \right)$$ Note: Eq. 10 omits the $1/m$ constant, which we will add in the revision.
>
>     + **Remark.** Another approach to derive Eq. 10: it is known that Wasserstein geodesics in the Euclidean space under the L2 distance metric are identical to Wasserstein barycenters (between the source and target distributions), which also gives rise to Eq. 10 (see Remark 9.8 of [4]).
>
> 2. **The experiment is insufficient**. Please see item-1 & item-2 in our [common response thread](https://openreview.net/forum?id=E1_fqDe3YIC&noteId=WPibEbBSzv) above for our new experiments on more datasets (CoverType, Color-Shift MNIST, and Office-Home). Also, we compare our GOAT with a new intermediate domain generation algorithm, CoVi [3] (which achieves state-of-the-art performance in UDA benchmarks), in the Portraits dataset, and provide the results in item-4 of our [common response thread](https://openreview.net/forum?id=E1_fqDe3YIC&noteId=6999nNFvnx) above.
>
> 3. **Visualization of Generated Intermediate Domains (e.g., Fig. 2).** Fig. 2 is an illustration to give readers an intuitive understanding (please see the “Feature Space vs. Input Space” paragraph in Sec. 3.3 on Page 6), while in practice, we do not directly generate intermediate domain data in the input space.
>
>     The intermediate domain generation of GOAT is performed in a latent feature space. To generate Fig. 2, we used a VAE since it has a decoder that can generate images from latent features. However, in practice, we find VAE does not work well (i.e., empirical performance is not good), thus we use CNN instead for our experiments in Sec. 4.
>
>     Since CNN has no decoder, we cannot convert the latent features of generated intermediate domains to images for visualization. That is why we do not provide more visualizations in addition to Fig. 2.

---

> > ### Author Response · Authors · 2022-11-12
> > **Response to Reviewer 3XRG (2/2)**
> >
> > 4. **When There are No Intermediate Domains Available (#Given Domains = 0)**. Please see item-3 in our [common response thread](https://openreview.net/forum?id=E1_fqDe3YIC&noteId=WPibEbBSzv) above for a detailed answer to this question.
> >
> > 5. **Comparison with Other Intermediate Domain Generation Algorithms**. Following your suggestion, we compare our GOAT with CoVi [3] (the newest and best-performing UDA algorithm across [1][2][3] that you pointed out) in the Portraits dataset. Please the experimental results in item-4 of our [common response thread](https://openreview.net/forum?id=E1_fqDe3YIC&noteId=6999nNFvnx) above. In summary, our GOAT outperforms CoVi in the Portraits dataset, which indicates that our algorithm is indeed powerful at i) generating intermediate domains useful for gradual domain adaptation and ii) leveraging given intermediate domains.
> >
> > **References**
> >
> > [1] Gong R, Li W, Chen Y, et al. Dlow: Domain flow for adaptation and generalization. CVPR 2019
> >
> > [2] Na J, Jung H, Chang H J, et al. Fixbi: Bridging domain spaces for unsupervised domain adaptation. CVPR 2021
> >
> > [3] Na J, Han D, Chang H J, et al. Contrastive Vicinal Space for Unsupervised Domain Adaptation. ECCV 2022
> >
> > [4] Peyré G, Cuturi M. Computational optimal transport. arXiv:1803.00567

---

### Official Review · Reviewer_tuhk · 2022-10-25

**Confidence:** 4
**Correctness:** 3
**Technical Novelty And Significance:** 2
**Empirical Novelty And Significance:** 2
**Recommendation:** 6

**Clarity, Quality, Novelty And Reproducibility:**


The Idea of generating intermediate domains, in my opinion, is good but requires more verification by conducting sufficient experiments.

The proposed method is verified on two datasets and compared with only one existing method, which seems insufficient.


**Strength And Weaknesses:**

*Positive points

The task, gradual domain adaptation, which the authors focus on, is truly important and practical in reality since it mitigates the limitation of UDA whenever the distribution shift between the source and the target is large.

The proposed GOAT adapts a progressive manner for shifting between different distributions along Wasserstein geodesic, which is guaranteed by a theory from wang et al[1].

The visualization results in figure.2 are intuitive for the semantic meaning of the generated intermediate domains, which helps readers to understand this paper better.


*Negative points

How to decide the number of generated domains of the proposed method? More discussions are needed.

1. In table 1, it is observed that the performance of GOAT suffers performance degradation with a large # Generated domains (i.e., # Given Domains=4, # Generated domains=4). Besides, when given 3 domains GOAT achieves the best performance with 2 generated domains, why is it the case? Why more intermediate domains would not boost the performance of GOAT?

2. But in Figure 4 (b), with the increase of generated domains, the test accuracy tends to increase, including Random, Uniform and OT settings. It seems that the contradictions in the experimental results.

It is necessary for the authors to provide more comprehensive and fair results.

1. The experiments are only conducted on two datasets (4 datasets are used in [1], 3 datasets are used in [2]), I would like to see the performance of the proposed GOAT in more datasets for further verification of the effectiveness of GOAT.

2. This paper only considers one baseline (GST, published in 2 years ago) for the comparison experiments. More recent state-of-the-art such as [1, 2] should be considered.

**Reference

[1] Understanding gradual domain adaptation: Improved analysis, optimal path and beyond. In ICML 2022.

[2] Kumar, A., Ma, T., & Liang, P. (2020, November). Understanding self-training for gradual domain adaptation. In ICML 2020.


**Summary Of The Paper:**

This paper aims to address the task of gradual domain adaptation (GDA) which aims to address large distribution shifts via adapting models along a sequence of intermediate domains. To overcome domain shifts (especially when the given intermediate domains are missing or scarce), the authors propose a generative gradual domain adaptation with optimal transport (GOAT). Specifically, the authors start by generating intermediate domains along the Wasserstein geodesic, then adapt the source-trained classifier to target by considering the intermediate domains. Experimental results demonstrate the effectiveness of the proposed GOAT. However, I still have concerns about this paper. My detailed comments are as follows.

**Summary Of The Review:**

The idea of generating intermediate domains for unsupervised domain adaptation is interesting to me. Moreover, the proposed GOAT is backed by a theory guarantee. However, the datasets and baselines for verifying the GOAT are insufficient for me. Thus, I recommend the weak rejection of this paper. If the authors convince me of the effectiveness of the GOAT, I would be glad to change the reviews.

---

> ### Author Response · Authors · 2022-11-12
> **Response to Reviewer tuhk**
>
> 1. **How to decide the number of generated domains?** Empirically, the number of generated domains is a hyperparameter that can be determined by a standard hyperparameter search, such as using a validation target set. Specifically, one can use a subset of the target set with highly confident pseudo-labels as a validation set. Then, one can evaluate the performance with different numbers of generated domains on the validation set and empirically select the optimal. In the paper (Table 1 & 2, Fig. 4b), we showed the performance of GOAT with different generated domains to show the effect of this hyper-parameter.
>
>     On the other hand, Wang et al. [1] prove the existence of a theoretically optimal number of intermediate domains (see Eq. 20 of [1]), and that also applies to our generated domains – however, the theoretical formula of [1] involves some unknown constants (that are dependent on the dataset and models, etc.), so we have to empirically determine the number of generated domains as a hyper-parameter in practice.
>
> 2. **Why more intermediate domains would not boost the performance of GOAT?** As we stated in the paper, [1] proves that more intermediate domains may not improve the performance – more accurately, there exists an optimal number of intermediate domains such that more domains beyond that will hurt the performance. [1] empirically verifies this theoretical prediction over four datasets. The reason behind that is the estimation error in each intermediate domain gets accumulated until it dominates. The conclusion of [1] also applies to our setting, no matter if the intermediate domains are ground-truth or generated, so it also happens in our case that more intermediate domains do not always boost the performance of GOAT.
>
> 3. **In Figure 4 (b), with the increase of generated domains, the test accuracy tends to increase, including Random, Uniform and OT settings.** The test accuracy does not necessarily increase with more generated domains. We show this by providing the results for more generated domains to extend Fig 4(b) -- please see the table below for details (rows correspond to transport plans, and columns stand for #Generated Domains). As we can see, after the generated domains are larger than 4, the performance does not increase for all methods, and degrades severely for random and uniform. This demonstrates that using OT to estimate the Wasserstein geodesic is indeed a vital step of our GOAT algorithm.
>
>     | Transport Plan \ # Generated Domains | 1          | 2          | 3          | 4          | 5          | 6          |
>     |---------------------|------------|------------|------------|------------|------------|------------|
>     | Random     | 55.3 ± 3.1 | 62.5 ± 2.6 | 62.0 ± 1.3 | **63.6 ± 2.8** | 54.7 ± 4.6 | 52.3 ± 2.0 |
>     | Uniform    | 57.3 ± 3.6 | 62.4 ± 2.8 | 62.4 ± 1.4 | **66.1 ± 1.6** | 59.9 ± 3.1 | 58.2 ± 3.0 |
>     | OT | 68.0 ± 1.4 | 67.0 ± 2.2 | 68.1 ± 2.2 | **70.3 ± 2.3** | 68.5 ± 2.5 | 68.8 ± 3.6 |
>     | Oracle | 67.9 ± 3.7 | 69.5 ± 1.9 | **71.4 ± 1.9** | **71.1 ± 1.4** | 68.9 ± 2.3 | 69.5 ± 3.1 |
>
> 4. **More datasets.** Please see item-1 in our [common response thread](https://openreview.net/forum?id=E1_fqDe3YIC&noteId=WPibEbBSzv) above for our new experiments on two datasets (CoverType & Color-Shift MNIST) that you suggested.
>
> 5. **Comparison with Newer/Improved Version of GST.** The GST [2] algorithm was indeed published two years ago, but there is actually no more improved version of GST so far. In fact, The work [2] that you pointed out is the original GST work, and [1] does not improve GST – [1] just runs the vanilla GST in cases of more intermediate domains to verify its theoretical prediction (“more intermediate domains may hurt the performance of GST”).
>
>     On the other hand, Reviewer 3XRG mentioned a new unsupervised domain adaptation (UDA) algorithm, CoVi [3] (achieving state-of-the-art performance on UDA benchmarks), which _generates intermediate domain_ data via MixUp and performs domain adaptation by utilizing the generated data. Since CoVi also performs _intermediate domain generation_ (though in a different way from ours), we conducted experiments to compare our GOAT with CoVi [3] in the Portraits dataset, showing a superior performance of GOAT compared with CoVi. Please see item-4 in our [common response thread](https://openreview.net/forum?id=E1_fqDe3YIC&noteId=6999nNFvnx) for detailed results of this experiment.
>
> **References**
>
> [1] Wang, H., Li, B. & Zhao, H. Understanding gradual domain adaptation: Improved analysis, optimal path and beyond. In ICML 2022.
>
> [2] Kumar, A., Ma, T., & Liang, P. Understanding self-training for gradual domain adaptation. In ICML 2020.
>
> [3] Na, J., Han, D., Chang, H. J., et al. Contrastive Vicinal Space for Unsupervised Domain Adaptation. In ECCV 2022.

---

> > ### Comment · Reviewer_tuhk · 2022-12-04
> > **Feedback to response**
> >
> > I would like to thank the authors for their detailed responses. My main concerns on the empirical side have been addressed now.

---

> > > ### Author Response · Authors · 2022-12-04
> > > **Thank you for the feedback**
> > >
> > > Dear Reviewer tuhk,
> > >
> > > We are glad that we have addressed your concerns! Thanks for your feedback.
> > >
> > > Sincerely,
> > >
> > > Authors of Paper3681

---

### Official Review · Reviewer_SoJb · 2022-10-28

**Confidence:** 4
**Correctness:** 4
**Technical Novelty And Significance:** 3
**Empirical Novelty And Significance:** 2
**Recommendation:** 8

**Clarity, Quality, Novelty And Reproducibility:**

I think the paper is well written and clearly presented. As a reader I would have preferred a higher-level, intuitive description of the algorithm before being presented with equations such as 4 and 5 (especially given that many of the details of equations 4 and 5 don't seem necessary to understand the main idea). However, this is partly a matter of taste.

One minor issue: I appreciate the fact that confidence intervals were given for the numbers in table 1 and 2. However, it seems to me that there are several case in which multiple columns should be bolded, because they are within margin of error (e.g., between 74.0±1.3 and 74.2±2.5 it makes little sense to bold the latter as being the "best").

The method seems novel, although I would argue that it seems a relatively straightforward idea given the existing theory and algorithms out there.

The models used are simple and clearly described in the appendix, so I imagine this work should be straightforward to reproduce.

**Strength And Weaknesses:**

I like this paper.

The idea seems well-grounded in theory. It seems to me that the authors are minimizing a lower bound though (the right-hand side of equation 6). I am also curious to know what cost function ($c$ in proposition 2). Is it L1 or (squared) L2-distance? Did the authors consider evaluating these different cost functions?

I appreciate the simplicity of the resulting algorithm: Pick a feature space to interpolate in, map both the source and target domain to this space, solve the optimal transport problem, and now just self-train your network as you slowly walk this path. This makes a lot of intuitive sense from the perspective of neural networks learning smooth "manifolds" of data, and GOAT exploits this well.

Experimentally, one concern I have is that GOAT does not seem very effective when there are no intermediate domains available (which seems to me the most common setting in practice). In particular, in table 1 the best approach for going from 0 to 45 degrees rotated MNIST seems to be to not generate any intermediate domains, and the gain for portraits is within margin of error. On the other hand, these tasks seem small and artificial enough (especially rotated MNIST) that I don't want to draw too strong conclusions from it.

Since GOAT does not require any intermediate domains, I would love to see GOAT evaluated using a few more domain adaptation datasets (e.g., the WILDS 2.0 dataset). This would give a much clearer signal as to how this algorithm performs in practice on real-world datasets. Is there a reason the authors did not do this? Does the EMD solver not scale to this size?

**Summary Of The Paper:**

This paper tackles the domain adaptation problem, i.e., transferring a model trained on a source domain to a new target domain for which unlabeled data is available. It does this using self-training (pseudo-labeling, noisy student) where the source-trained model is used to generate labels for the target domain. The paper builds on theoretical insights which suggest self-training works best when the input data shifts gradually from the source to target domain (gradual self-training). This theory suggests that the ideal case would involve walking the shortest path between all the joint data distributions of the intermediate domains.

The authors transform this theoretical insight into a practical algorithm by:

* making a covariate shift assumption which allows them to operate in a latent space, rather than the joint distribution of the data
* generating intermediate domains that lie close to the Wasserstein geodesic between the source and target domain
* adding a regularization term to the optimal transport problem to reduce computational complexity, in combination with thresholding the values in the transport plan to minimize space complexity

**Summary Of The Review:**

I think this is a good paper. It's main shortcoming is the experimental evaluation, which is limited to very small networks on small and artificial datasets. If such experiments are added, I would be able to support acceptance of this paper more strongly.

---

> ### Author Response · Authors · 2022-11-12
> **Response to Reviewer SoJb**
>
> 1. **Right-Hand Side of Eq. (6).** Eq. 6 just demonstrates that the path length of intermediate domains (left-hand side) is lower bounded by the Wasserstein geodesic length (right-hand side). This lower bound (right-hand side) is deterministic (determined by the source and target data distributions, $\mu_0$ and $\mu_T$), and our algorithm does not try to minimize it. Instead, we are trying to minimize the left-hand side of Eq. 6 (path length of intermediate domains) — Proposition 2 shows that this path length (left-hand side) reaches the lower bound (right-hand side) as the intermediate domains fall along the Wasserstein geodesic — this theoretical observation leads us to propose our algorithm, GOAT, which directly generates intermediate domains along the Wasserstein geodesic so that the path length is minimized (i.e., equal to the right-hand side of Eq. 6).
>
> 2. **Cost Function Choice in Proposition 2.** Proposition 2 does not specify the cost function, which only needs to satisfy $c: \mathcal{X} \times \mathcal{X} \mapsto [0,\infty)$. For example, L1, L2 or any other Lp distance metric satisfies this condition. In this paper, we adopt the L2 distance in our implementation, since it is the most common choice of the cost function for computational optimal transport.
>
> 3. **When There are No Intermediate Domains Available.** Please see item-3 in our [common response thread](https://openreview.net/forum?id=E1_fqDe3YIC&noteId=WPibEbBSzv) above for a detailed answer to this question.
>
> 4. **More Domain Adaptation Datasets.** Please see item-1 & item-2 in our [common response thread](https://openreview.net/forum?id=E1_fqDe3YIC&noteId=WPibEbBSzv) above for our new experiments on more datasets (CoverType, Color-Shift MNIST, and Office-Home).
>
> 5. **Scalability of Optimal Transport Solver.** As for the EMD solver, its theoretical complexity is nearly $O(n^2)$ when used with entropy regularization (will be $O(n^3 \log n)$ without the entropy regularization), where $n$ is the number of data in the source/target dataset (we assume they are of the size here for simplicity). We adopted the entropy regularization in our implementation, and we observed that the EMD solver takes ~20min for a dataset of n=50K on a 48-core CPU. Therefore, for large domain adaptation datasets such as VisDA-17 (152K source and 280K target data), we estimate that the EMD solver may take ~6hr. For larger datasets, some computation optimal transport strategies, such as mini-batch optimal transport [1], can be applied to accelerate the optimal transport computation.
>
> 6. **Multiple Columns Should be Bolded.** Thank you for the suggestion. We will modify them accordingly in our updated version later.
>
>
> **References**
>
> [1] Nguyen et al. Improving Mini-batch Optimal Transport via Partial Transportation. In ICML 2022.

---

### Author Response · Authors · 2022-11-12
**Response to Common Requests/Questions of Reviewers (1/2)**

1. **More Datasets (SoJb, Tuhk and d4Li).** As requested by reviewer SoJb, Tuhk and d4Li, we have conducted experiments in two more datasets of gradual domain adaptation (GDA), CoverType (used in [1][2]) and Color-Shift MNIST (used in [1]). Similar to Table 1 & 2 in our paper, the rows ("#Given")stand for _the number of **given** (ground-truth) intermediate domains_ and columns ("#Generated") represent _the number of **generated** domains (between each pair of adjacent ground-truth domains) for GOAT_. As #Generated=0, GOAT is reduced to gradual self-training (GST) [2]. Each experiment is run over 5 random seeds, and we report its mean accuracy with 95% confidence interval.

    (a) **CoverType**
    |  #Given \ #Generated | 0 (GST)    | 1  | 2  | 3 | 4 |
    |---|----|---|--|------------|------------|
    | 0 | 63.0 ± 2.3 | 64.2 ± 2.2 | 65.0 ± 2.4 | 66.2 ± 2.1 | **66.5 ± 2.0** |
    | 1 | 65.9 ± 2.1 | 68.5 ± 2.0 | 68.4 ± 1.5 | **69.1 ± 1.5** | **69.1 ± 1.5** |
    | 2 | 66.9 ± 1.4 | 68.9 ± 1.6 | 68.4 ± 2.1 | 69.3 ± 1.1 | **69.8 ± 1.4** |
    | 3 | 66.9 ± 1.3 | 68.3 ± 1.4 | **69.9 ± 1.8** | 68.0 ± 1.5 | 68.8 ± 1.1 |
    | 4 | 67.7 ± 1.7 | **69.6 ± 2.1** | 68.1 ± 2.0 | **69.7 ± 1.2** | 69.4 ± 2.0 |

    (b) **Color-Shift MNIST**
    | #Given \ #Generated  | 0 (GST)    | 1          | 2          | 3          | 4          |
    |---|--|---|------------|------------|------------|
    | 0 | 40.5 ± 5.5 | 54.4 ± 6.9 | 63.2 ± 4.1 | 75.7 ± 3.8 | **79.1 ± 3.0** |
    | 1 | 54.2 ± 5.9 | 74.7 ± 5.3 | 79.5 ± 2.9 | 79.3 ± 3.4 | **85.3 ± 3.8** |
    | 2 | 67.6 ± 4.8 | 78.3 ± 3.4 | 84.8 ± 2.5 | 89.0 ± 1.5 | **90.3 ± 1.4** |
    | 3 | 73.9 ± 7.6 | 80.9 ± 6.9 | 87.4 ± 4.2 | **90.7 ± 2.3** | 90.4 ± 1.5 |
    | 4 | 77.4 ± 7.2 | 84.4 ± 4.6 | **91.8 ± 1.8** | 91.0 ± 1.8 | 91.3 ± 1.2 |

    Generally speaking, the performance trend above is similar to the Rotated-MNIST and Portraits. Namely, the performance of GOAT (with appropriate choices of #generated domains) increases monotonically with more oracle intermediate domains. Moreover, GOAT consistently outperforms gradual self-training (GST). Also, as one can see from some rows, more generated intermediate domains do not necessarily lead to better performance.

2. **UDA Benchmark (SoJb, 3XRG).** As suggested by reviewer SoJb and 3XRG, we conducted an experiment for GOAT in an unsupervised domain adaptation (UDA) benchmark. Specifically, we use Office-Home, a common UDA dataset, and compare GOAT with self-training, since our GOAT is a self-training-based algorithm. Due to the time limit, we cannot complete the experiments of training large modern networks (e.g., ResNet-50 and beyond), which require lots of tuning and hyperparameter searches, etc. Hence, we adopt a CLIP-pretrained ViT/B-32 [3] network as a fixed backbone encoder (which has been shown to be a good feature extractor for a wide range of datasets [3]) and train an MLP adapter with our GOAT algorithm on top of this encoder. In each domain adaptation task, we generate 2 intermediate domains for GOAT, and report the average accuracy over 5 runs for self-training and GOAT.

    |      | Ar->Cl | Ar->Pr | Ar->Rw | Cl->Ar | Cl->Pr | Cl->Rw | Pr->Ar | Pr->Cl | Pr->Rw | Rw->Ar | Rw->Cl | Rw->Pr | Average  |
    |------|--------|--------|--------|--------|--------|--------|--------|--------|--------|--------|--------|--------|------|
    | Self-Training  | 58.4   | 73.8   | 79.0     | 65.1   | 77.0     | 76.7   | 62.6   | 58.2   | 80.2   | **71.6**   | 61.5   | **86.7**   | 70.9 |
    | GOAT | **60.9**   | **78.1**   | **80.3**   | **67.6**   | **80.0**     | **78.7**   | **65.1**   | **61.5**   | **81.0**     | **71.7**   | **64.5**   | **86.2**   | **73.0**   |

    As we can see, GOAT consistently outperforms self-training across UDA tasks in Office-Home. However, we want to emphasize that although GOAT can work in the UDA setting, the focus of this work is still to show how it leverages the pattern of gradual distribution shift, which is often not a property of distribution shift in traditional UDA benchmarks.


3. **The Case of No Ground-Truth Intermediate Domain (SoJb, 3XRG, d4Li).** Without any ground-truth intermediate domain, the problem is reduced to the classical setting of unsupervised domain adaptation (UDA), which is not the focus of this paper. Instead, this paper focuses on gradual domain adaptation (GDA), which typically comes with some intermediate domain data (may be scarce). In this regime, UDA methods are not suitable since they are not designed to leverage intermediate domains. Our GOAT mainly aims to improve standard GDA when the given intermediate domains are scarce. As we show in Fig. 3(a) of the paper, without any ground-truth intermediate domain, GOAT may not estimate the underlying ground-truth geodesic accurately as the distribution shift is very large. In this case, some intermediate domain(s) are necessary to guide GOAT to better estimate the geodesic, as illustrated by Fig. 3(b).

---

> ### Author Response · Authors · 2022-11-12
> **Response to Common Requests/Questions of Reviewers (2/2)**
>
> 4. **Comparison with Other Intermediate Domain Generation Algorithm (Tuhk, 3XRG).** Following the suggestion of Reviewer 3XRG, we compare our GOAT with CoVi [6] (the newest and best-performing unsupervised domain algorithm across [4][5][6] that Reviewer 3XRG mentioned) in the **Portraits** dataset. CoVi generates intermediate domain data via MixUp and performs domain adaptation by utilizing the generated data, and it archives state-of-the-art performance on modern unsupervised domain adaptation (UDA) benchmarks. To enable CoVi to leverage intermediate domains, we run it in a similar way to gradual self-training (GST): pseudo-labeling intermediate domain data and iteratively running the algorithm over the sequence of domains. Here we show our empirical results on the Portraits dataset.
>
>     | # Given Domains | GST [2]       | CoVi [6]   | GOAT (Ours)  |
>     |-----------------|------------|------------|--------------|
>     | 0               | 73.3 ± 1.3 | 73.7 ± 3.5 | **74.2 ± 2.5​​** |
>     | 1               | 74.5 ± 1.6 | 75.3 ± 1.8 | **76.8 ± 1.5**   |
>     | 2               | 77.0 ± 1.3 | **79.8 ± 3.0** | **79.9 ± 1.2**   |
>     | 3               | 80.7 ± 2.3 | **82.3 ± 1.4** | **82.3 ± 1.3**   |
>     | 4               | 82.0 ± 1.4 | 83.1 ± 1.9 | **83.6 ± 1.5**   |
>
>     + **Code.** We adopt the official code of CoVi (https://github.com/NaJaeMin92/CoVi), and merge it into our codebase. We have already updated our Supplementary Material with our new codebase that includes CoVi.
>
>     + **Implementation.** For a fair comparison, we fix the network structure CoVi to be the same as the one used in our paper, and also adopt the same training recipe (Adam optimizer with weight decay, batch size = 128). Same as gradual self-training (GST) & GOAT, we run CoVi over at least 5 random seeds and report the average accuracy with 95% confidence interval.
>
>     + **Results.** In the above experiment on Portraits, the performance of GOAT is comparable to or outperforms CoVi across all numbers of given domains (0,1,...,4). That indicates that our algorithm is indeed powerful at i) generating intermediate domains useful for gradual domain adaptation and ii) leveraging given (ground-truth) intermediate domains.
>
>     + **Remark.** Even though our GOAT outperforms CoVi in the above experiment, that does not imply GOAT can also outperform CoVi on modern UDA benchmarks, since the type of distribution shift of UDA datasets is quite different from that of the GDA datasets (e.g., Portraits and rotated MNIST).
>
>
>
>
> **References**
>
> [1] Wang, H., Li, B. & Zhao, H. Understanding gradual domain adaptation: Improved analysis, optimal path and beyond. In ICML 2022.
>
> [2] Kumar, A., Ma, T., & Liang, P. Understanding self-training for gradual domain adaptation. In ICML 2020.
>
> [3] Radford, Alec, et al. Learning transferable visual models from natural language supervision. In ICML 2021.
>
> [4] Gong R, Li W, Chen Y, et al. Dlow: Domain flow for adaptation and generalization. CVPR 2019
>
> [5] Na J, Jung H, Chang H J, et al. Fixbi: Bridging domain spaces for unsupervised domain adaptation. CVPR 2021
>
> [6] Na J, Han D, Chang H J, et al. Contrastive Vicinal Space for Unsupervised Domain Adaptation. ECCV 2022

---

### Decision · Program_Chairs · 2023-01-20

**Decision:**

Reject

**Justification For Why Not Higher Score:**

we would have expected
* clarification of the theoretical contributions : it should be made clear that most of the theoretical parts are
well known results in OT and relations with existing works should be discussed and analyzed in details.

* and more experimental results with gradual DA datasets. Since gradual DA is an important problem, we believe
that there are more datasets/problems on which it can applied.


**Justification For Why Not Lower Score:**

N/A

**Metareview: Summary, Strengths And Weaknesses:**

The paper  proposes to use a wasserstein distance based method for generating intermediate domains for gradual
domain adaptation (DA). The proposed method is evaluated on a toy(ish) problem (MNIST) and one gradual
domain adaptation problem. From my point of view, the contribution of this paper is essentially
about applying wasserstein interpolating measures equation to gradual domain adaptation


After discussion, it becomes clear that the theoretical/methodological contribution is a bit weak and the contribution stands mostly in applying well known "interpolating measure" notions in optimal transport to DA. As such, we would have expected the experimental analyses to be stronger with more real-world gradual DA datasets. Hence, at this point, we believe that the paper deserves some further improvements before acceptance.

---

> ### Author Response · Authors · 2023-01-25
> **Response to Meta-Review: Additional Datasets of Gradual Domain Adaptation are Already Included in Rebuttal**
>
> We are grateful for the dedication of the reviewers and area chairs. However, we noticed that the meta-review overlooked the supplementary experiments we submitted during the rebuttal phase. We would like to emphasize that we have included experiments on two additional gradual domain adaptation datasets, CoverType and Color-Shift MNIST, in our previous [author response](https://openreview.net/forum?id=E1_fqDe3YIC&noteId=WPibEbBSzv) below.